# The Gut Microbiome’s Influence on Incretins and Impact on Blood Glucose Control

**DOI:** 10.3390/biomedicines12122719

**Published:** 2024-11-27

**Authors:** Ajay S. Dulai, Mildred Min, Raja K. Sivamani

**Affiliations:** 1Integrative Research Institute, Sacramento, CA 95819, USA; 2Integrative Skin Science and Research, Sacramento, CA 95815, USA; 3College of Medicine, California Northstate University, Elk Grove, CA 95757, USA; 4Pacific Skin Institute, Sacramento, CA 95815, USA; 5Department of Dermatology, University of California-Davis, Sacramento, CA 95616, USA

**Keywords:** microbiome, incretins, *Akkermansia*, probiotics, obesity, diabetes, glucose

## Abstract

Obesity and type 2 diabetes mellitus (T2DM) have been increasing in prevalence, causing complications and strain on our healthcare systems. Notably, gut dysbiosis is implicated as a contributing factor in obesity, T2DM, and chronic inflammatory diseases. A pharmacology exists which modulates the incretin pathway to improve glucose control; this has proven to be beneficial in patients with obesity and T2DM. However, it is unclear how the gut microbiome may regulate insulin resistance, glucose control, and metabolic health. In this narrative review, we aim to discuss how the gut microbiome can modulate incretin pathways and related mechanisms to control glucose. To investigate this, Google Scholar and PubMed databases were searched using key terms and phrases related to the microbiome and its effects on insulin and glucose control. Emerging research has shown that several bacteria, such as *Akkermansia* and MN-Gup, have GLP-1-agonistic properties capable of reducing hyperglycemia. While more human research is needed to prove clinical benefit and identify long-term implications on health, the usage of pre-, pro-, and postbiotics has the potential to improve glucose control.

## 1. Introduction

In 2017, a total of 462 million individuals globally were living with type 2 diabetes mellitus (T2DM), resulting in an increasing trend in diabetic-related complications and economic burden on healthcare systems [1]. In the United States, the annual cost of T2DM management exceeded $327 billion in 2017 [2]. In developing countries, under-resourced healthcare systems and increasing patient numbers lead to significant financial strain. Unmanaged or poorly controlled T2DM also lead to severe health complications. Cardiovascular disease is the most common, significantly increasing the risk for heart attacks and strokes. Other complications include neuropathy, nephropathy, which may progress to kidney failure, and retinopathy, which is the leading cause of blindness in working-age adults [3]. Together, these complications increase disability rates and reduce quality of life, emphasizing the need for improved management and prevention strategies.

The traditional management of T2DM relies on various pharmacological agents, each targeting a different component related to glucose regulation. First-line therapy includes biguanides, primarily metformin, which reduce hepatic glucose production and improve insulin sensitivity in peripheral tissues [4]. Although metformin is generally well tolerated, side effects include nausea and diarrhea, which increases the risk for vitamin B12 deficiency with long-term use [5]. Sulfonylureas, like glimepiride and glyburide, stimulate insulin release from pancreatic beta cells; however, they carry the risk of hypoglycemia and weight gain, especially among older adults [6]. Sodium–glucose co-transporter-2 (SGLT-2) inhibitors, including empagliflozin and dapagliflozin, work by preventing glucose reabsorption in the kidneys, promoting its excretion in the urine [7]. Despite the variety of available medications, several limitations remain. One challenge is the risk of treatment failure over time, as some patients may develop resistance to drugs like sulfonylureas. These medications also carry side effects, which can reduce patient adherence. While current treatment guidelines provide a general framework, individual responses may vary widely based on genetics, lifestyle, and disease progression. These limitations underscore the challenge of managing T2DM long-term; therefore, there is a pressing need for new strategies that address the underlying mechanisms of glucose dysregulation.

One potential strategy may include the gut microbiome. The gut microbiome consists of the bacteria, yeast, and fungus found within the human gastrointestinal system. The microbiome is essential for normal functioning of the gastrointestinal system, and its dysregulation is implicated in numerous chronic diseases [8]. Notably, research suggests that gut dysbiosis is associated with insulin resistance, obesity, and T2DM [9,10]. Thus, modulating the gut microbiome may promote better glucose control. Currently, there is no treatment targeting the gut microbiome to modulate glucose control [11]. Additionally, obesity is strongly associated with T2DM, which affected 650 million adults in 2022 [12]. Insulin resistance is a common underlying feature of T2DM and obesity which manifests from decreased insulin-stimulated glucose secretion and metabolism, and impaired suppression of hepatic glucose production as well as contribution from the gut microbiota [13,14]. An overlap in the pharmacological management of T2DM and obesity exists with medications targeting the production and regulation of incretins.

Incretins are specialized peptides that are released from the gastrointestinal tract in response to nutrient intake, playing a crucial role in amplifying post-prandial insulin secretion [15]. The two incretin hormones are glucose-dependent insulinotropic polypeptide (GIP), produced by the K cells in the duodenum and upper jejunum; and glucagon-like peptide-1 (GLP-1), produced by the L cells in the small and large intestines [16]. Together, these incretin hormones work synergistically to enhance insulin secretion, with their combined effect estimated to contribute approximately 74% to post-prandial insulin release [17]. The proper function of incretin hormones is critical for maintaining glucose homeostasis, and any dysregulation in their secretion or action can lead to significant metabolic issues. Such dysregulation of incretin secretion and its downstream effects on pancreatic beta cell insulin secretion is implicated in the pathophysiology of both T2DM and obesity [18]. Specifically, in T2DM, the normal physiological response of incretins is compromised, leading to inadequate glucose control. While GIP is estimated to be the more potent insulinotropic hormone, the GLP-1 pathway is conserved to a further degree in T2DM [19]. This conservation suggests that despite the challenges in incretin function associated with T2DM, the GLP-1 pathway still holds potential for therapeutic intervention. Figure 1 provides a visual summary of how the incretin pathway modulates insulin secretion.

Incretin-based therapies have revolutionized the treatment of T2DM and obesity since their development, with the discovery of GLP-1 receptor agonists playing a pivotal role. GLP-1 agonists mimic the effects of endogenous GLP-1 by stimulating insulin release, inhibiting glucagon secretion, and delaying gastric emptying [23]. GLP-1 agonists include semaglutide, exenatide, and liraglutide, which quickly became popular due to their ability to promote weight loss, making them effective in also managing hyperglycemia and obesity. Common side effects include nausea, vomiting, and gastrointestinal discomfort, which may pose challenges for patient adherence [24]. Among GLP-1 agonists, shorter-acting agonists target postprandial hyperglycemia more effectively, while longer-acting agonists primarily influence fasting glucose levels [25]. Building on GLP-1 agonists are dual GIP/GLP-1 receptor agonists, such as tirzepatide [26]. Dual agonists offer enhanced glucose control and more significant weight loss due to the additive effects of GIP and GLP-1. In clinical trials, tirzepatide has led to more than 10% body weight loss, and reductions in HbA1c (maximum reduction of 2.24%) [26,27,28]. Tirzepatide is also currently being investigated for its benefits on cardiovascular event risk [27]. However, these agents have a higher risk of gastrointestinal side effects and more research is needed to understand their long-term safety profiles. Another approach involves the inhibition of dipeptidyl peptidase 4 (DPP-4), the enzyme responsible for the breakdown of incretins, with medications such as sitagliptin [29]. Overall, incretin-based therapies continue to push the boundaries of diabetes treatment; however, ongoing studies are necessary to refine dosing, minimize side effects, and determine long-term sustainability of these interventions.

The ability of the gut microbiome to produce a diverse array of metabolites that can modulate incretin hormones like GLP-1 and GIP has garnered increasing interest. By enhancing or inhibiting the secretion of these incretins, gut microbiota may play a role in amplifying post-prandial insulin release, reducing hyperglycemia, and improving glucose control. For example, butyrate, a short-chain fatty acid (SCFA) has been shown to positively influence insulin sensitivity and reduce inflammation, which are key factors in the pathogenesis of insulin resistance and T2DM [30]. Additionally, propionate, another SCFA, has been linked to reduced glucose production in the liver and enhanced insulin sensitivity [31]. Specific bacterial species, such as *Akkermansia muciniphila* and *Bifidobacterium*, are associated with improving metabolic health and increasing the abundance of SCFAs like butyrate and propionate. Notably, the increased abundance of these bacterial species has been inversely correlated with obesity and T2DM [32]. This review will delve into the various components of the gut microbiome that have demonstrated the ability to modulate glucose levels and insulin sensitivity. By exploring these interactions, we aim to provide an overview of how gut microbiome-targeted approaches may improve glycemic control and reduce the risk of diabetes-related complications.

## 2. Search Strategy

To gather comprehensive information on the microbiome’s influence on glucose control and insulin regulation, an extensive literature search was conducted using the databases Google Scholar and PubMed from inception to 14 June 2024. Our search was initiated by identifying key terms relevant to the relationship between the gut microbiome and metabolic health, specifically focusing on insulin secretion, glucose metabolism, and incretin modulation. Initial search phrases included combinations of key words such as “gut microbiome impact on insulin”, “*Akkermansia* effects on insulin sensitivity”, “microbiome modulation of GLP-1”, and “probiotics in diabetes treatment”. These keywords were refined based on the relevance of the results, and additional terms were added to explore related areas, including “role of short-chain fatty acids in glucose control” and “gut microbiota and type 2 diabetes mellitus”. This inclusion criteria were broad to capture a wide scope of evidence. We included in vitro studies, as these provided mechanistic insights for future clinical applications. We also include in vivo studies, primarily focusing on clinical trials conducted in humans or animal models, particularly mice, given their widespread use in metabolic research. The review process did not limit articles based on publication date but resulted in more recent studies, reflecting the emerging developments in the fields. Moreover, we conducted secondary searches by scanning the reference lists of relevant articles, identifying additional sources that may not have appeared in the initial database queries. Exclusion criteria included studies that lacked clear relevance to microbiome modulation, were solely focused on unrelated metabolic pathways, were not published in English, or were not published in a peer-reviewed journal.

### 2.1. Search Results

We included a total of 12 articles in this review that explore the impact of the gut microbiome on insulin regulation, glucose metabolism, and the modulation of incretin hormones. Out of the 12 studies, 4 were in vitro studies, where bacterial strains were examined in controlled environments outside of living organisms. The remaining 8 studies were in vivo studies conducted in animal models or humans, offering a broader understanding of how these bacterial populations behave within the gut microbiome and how they affect metabolic outcomes. One common feature across all studies was the use of 16s rRNA gene sequencing to analyze the microbial composition. Table 1 and Table 2 summarize the key findings of each study included in this review.

Several key proteins isolated from *Akkermansia* have demonstrated GLP-1 stimulating effects. The P5 protein isolated from pasteurized *A. muciniphila* can increase GLP-1 release from STC-1 cells via the free fatty acid receptor 2 (FFAR-2) [41]. The P9 protein, when isolated from *A. muciniphila* and incorporated into *L. lactis*, demonstrated upregulation of the GLP-1 biosynthesis genes GCG (1.63-fold) and PCSK1 (1.53-fold) [42]. Mice studies suggest that P9 functions through interaction with the intercellular adhesion molecule 2 (ICAM-2) and can induce GLP-1 secretion [39].

### 2.2. Lactobacillus and Bifidobacterium

*Lactobacillus* and *Bifidobacterium* isolated from healthy children were evaluated to determine their effects on metabolic control in STC-1 cells [43]. The cell-free extract (CFE) of *Lactobacillus rhamnosus GG* (LGG) and *Lactiplantibacillus pentosus* L-8 had the highest degree of α-glucosidase inhibitory activity, 24.4% and 23.8% respectively. Furthermore, the CFE of *Bifidobacteria longum* strains 6-2 and B-53 showed significantly higher α-glucosidase inhibitory activity compared to other strains (*p* < 0.05). It was determined that strains BB12, L-8, 6-2, and B-53 had the most potent α-glucosidase inhibitory activity. The cell-free supernatant (CFS) from LGG, L-8, and B-84 were found to have the highest DPP-IV inhibitory activity.

*Bifidobacterium animalis* subsp. *lactis* MN-Gup (MN-Gup) has been researched in mice models for its hypoglycemic effects. A study demonstrated that a 6-week course of MN-Gup supplementation in T2DM mice resulted in reduced fasting blood glucose and homeostasis model assessment-insulin resistance (HOMA-IR). Furthermore, the MN-Gup treatment group had increased acetate levels (*p* < 0.05) and GLP-1 levels (*p* < 0.05) [33]. The results showed that higher-dose MN-Gup had greater benefits on fasting glucose compared to the low-dose MN-Gup group (Table 1). Another study found that MN-Gup supplementation in healthy mice was protective against diet-induced weight gain in a dose-dependent manner [34]. MN-Gup additionally demonstrated protective effects against high fat diet induced increases in total cholesterol (TL), triglycerides (TGs), and low-density lipids (LDLs) [34]. The addition of fermented milk provided further benefits in increasing high-density lipid levels (HDLs). Further mice studies have concluded with similar results regarding the beneficial metabolic effects of MN-Gup when administered with fermented milk [35]. Results from this treatment included reduced TL, TG, and LDL (*p* < 0.05). Notably, the addition of Galacto-oligosaccharides (GOS) and xylo-oligosaccharides (XOS) contributed to a significant reduction in weight gain compared to the control group (*p* < 0.05).

*Bifidobacterium animalis* ssp. *lactis* 420 (B420) (10^9^ CFU/day) supplementation has been investigated with and without metformin, sitagliptin, and prebiotic polydextrose (PDX) in T2DM mice. Mice treated with B420 had better glucose tolerance (2140 mmol/L*min) when compared to vehicle (2340 mmol/L*min) (*p* = 0.002) [37]. While B420 did not demonstrate increased GLP-1 secretion, PDX was hypothesized to improve glucose control via stimulation of GLP-1.

### 2.3. S. epidermidis

In 2022, researchers screened bacterial strains in vitro for GLP-1 stimulatory effects. Results showed that 45 strains of *S. epidermidis* resulted in an increase in GLP-1, most notably the JA1 and JA8 strains [36]. The proposed peptide of interest, named Hld_Se_, is thought to stimulate GLP-1 secretion via a calcium-dependent reaction. Hld_Se_ when administered to in vitro enterocytes demonstrated increased GLP-1 production (*p* < 0.05) and increased serotonin production (*p* < 0.05) with no change to cell viability. While all strains of *S. epidermidis* expressed Hld_Se_, only the strains without GLP-1 stimulatory effects contained a single gene polymorphism in the *AgrA* gene. This suggests that normal functioning of the *AgrA* gene is essential for Hld_Se_ to stimulate GLP-1 (Figure 2). In vitro administration of Hld_Se_ resulted in a 3- to 4-fold increase in GLP-1 secretion. Further evaluation of JA1 demonstrated reduced food intake, body mass, adiposity, and fasting insulin when compared to the control.

### 2.4. Akkermansia

Several key proteins isolated from *Akkermansia* have demonstrated GLP-1-stimulating effects. The P5 protein isolated from pasteurized *A. muciniphila* can increase GLP-1 release from STC-1 cells via the free fatty acid receptor 2 (FFAR-2) [41]. The P9 protein, when isolated from *A. muciniphila* and incorporated into *L. lactis*, demonstrated upregulation of the GLP-1 biosynthesis genes GCG (1.63-fold) and PCSK1 (1.53-fold) [42]. Mice studies suggest that P9 functions through interaction with the intercellular adhesion molecule 2 (ICAM-2) and can induce GLP-1 secretion [39].

*Akkermansia* is known to be more abundant in the guts of healthy individuals when compared to diabetic and obese patients [39,40,41]. The effect that *Akkermansia* has on in-cretins provides insight into the abundance of it in the healthy gut and its potential for use in modulating GLP-1. However, there are few clinical studies evaluating supplementation with *Akkermansia* and evaluating the levels of incretins. Furthermore, research into metformin has shown that it increases the abundance of *A. muciniphila* in diet-induced obese mice and humans [44]. While in vitro studies have identified specific proteins from *Akkermansia* which can aid in insulin response, and human trials have demonstrated associations between *Akkermansia* and metabolic changes, clinical trials are needed to demonstrate that *Akkermansia* supplementation can induce incretin stimulation in animal models and humans.

### 2.5. Postbiotics

While prebiotics and probiotics have shown promising results, some bacterial products have demonstrated beneficial effects in vitro. Indole, a metabolite of dietary tryptophan, has demonstrated stimulatory effects on GLP-1 via inhibition of voltage-gated potassium channels in GLUTag cells [45]. While 1 mM of indole caused a rapid initial secretion of GLP-1 after 5 min, the total GLP-1 secretion after 240 min was 35% less than the control. Indole’s mechanism of action is theorized to inhibit voltage-gated potassium channels, leading to widening of the action potential and increased intracellular concentration of calcium in the GLUTag cell. The long-term inhibitory effect is hypothesized to be a result of decreased intracellular ATP generation. Further testing demonstrated similar results in mouse derived intestinal L-cells.

Indole is produced by bacteria which contain the tryptophanase gene named TnaA [46]. A variety of taxa express this gene, including *Clostridia*, *Bacteroide*, and *Gammaproteobacteria* [47]. Furthermore, indole can be produced by *E. coli* and degraded by *P. aeruginosa* [48]; however, more long-term studies are needed to determine if the net effect of indole is stimulatory or inhibitory on GLP-1 release.

There is limited research into the effects of the microbiome on GIP. One study identified that the administration of maltose and miglitol had stimulatory effects on GLP-1 but inhibitor effects on GIP in mice [40]. The suppression of GIP is theorized to be modulated by the microbiome, as antibiotic administration attenuated this response. Furthermore, the free fatty acid receptor 3 (FFAR3) is thought to be implicated in this process, as mice deficient in FFAR3 did not demonstrate GIP inhibition.

### 2.6. Pathogenic Species

Notably, the reduction in potentially pathogenic species can have beneficial effects on glucose control. In diet-induced obese mice, vancomycin and bacitracin depleted *Firmicutes* and *Bacteroidetes* in the gut. The metabolic results of antibiotic treatment include a 2.03-fold increase in active GLP-1, reduced systemic glucose intolerance, and reduced hyperinsulinemia [38].

## 3. Prebiotics and Postbiotics to Modulate the Microbiome

Apart from supplementation with live bacteria as probiotics, other strategies have included the use of prebiotics or postbiotics to increase the gut’s natural levels of incretin-stimulating bacteria. Prebiotics are defined as nutrients which are metabolized by our microbiome [49]. Oral supplementation with fructo-oligosaccharides (FOSs) has been shown to significantly increase the bacterial abundance of *A. muciniphila* and *Bifidobacterium* spp. in animal models [50,51,52]. Further research has shown that administration of polyphenol-rich cranberry extract in T2DM mice improved features of metabolic syndrome and increased the relative abundance of *Akkermansia* [53]. Direct supplementation with live probiotics has promise for increasing the relative abundance of beneficial strains, including SCFA producers [54]. Postbiotics involve the use of probiotic metabolites or pasteurized bacteria [55]. Pasteurized products from *Akkermansia* demonstrated GLP-1 agonistic benefits [41], suggesting that postbiotics can provide similar benefits to live probiotics. The mechanism of this action likely relies on gene uptake by live bacteria in the gut [56]. This is significant due to the higher cost and difficulty of manufacturing associated with live probiotics. Future studies will help delineate the effect of prebiotics and postbiotics in comparison to probiotics.

## 4. Limitations of Current Research

One of the issues that has emerged with clinical use of GLP-1 agonists is rebound weight gain when the medications are stopped. For example, those that stopped semaglutide regained two-thirds of their prior weight loss within the following year [57], suggesting that GLP-1 agonism needs to be paired with lifestyle changes to ensure long-term weight management. Effective adjunctive lifestyle changes include a diet which consists of a 500–750 calorie deficit [58], intermittent fasting [59], and regular weight-bearing exercise [60]. Adequate daily protein intake of 0.8 to 1.2 g/kg of body weight is also critical to preserve lean muscle mass [61]. For those with restrictions for weight-bearing exercise and protein intake, referral to ancillary services may be necessary. The role of adjuvant supplementation in maintaining weight loss is largely unknown but should be considered for future studies with GLP-1 agonists.

Many studies did not identify if total GLP-1 or only active GLP-1 was reported. While active GLP-1 is responsible for the endocrine incretin effect, the various isoforms of GLP-1 are hypothesized to have central neural stimulatory effects [62]. Research has recommended that the measurement of total GLP-1 can provide a better estimator of its cumulative effects [63]. Further research is needed to understand the affect the microbiome has on GIP. While GLP-1 and GIP are thought to have similar effects on insulin, they can also have differing effects throughout the body. GIP has been implicated with increased fat deposition, an increase in postprandial glucagon secretion, and promotion of bone formation, while GLP-1 inhibits postprandial glucagon secretion and inhibits bone resorption [20]. Analyzing incretins as a whole or purely GLP-1 in research inhibits our full understanding of these compounds.

Despite promising findings on the role of the microbiome in modulating glucose control, there are several limitations to discuss. First, the variability in gut microbiota composition between individuals poses a challenge. Factors such as diet, environment, genetics, and medication use influence microbial diversity, making it difficult to generalize findings or develop standardized therapies. This heterogeneity necessitates personalized approaches, which are still in their infancy due to the complexity of the microbiome and the difficulty in identifying consistent biomarkers for intervention.

Another limitation is the lack of long-term studies assessing the sustainability of microbiome-targeted interventions. While some studies suggest positive short-term outcomes, the long-term safety and efficacy of probiotic, prebiotic, and postbiotic supplementation remain unclear. For example, the sustainability of these effects after discontinuation of supplementation is unknown, mirroring concerns seen with pharmacological treatments, where rebound weight gain occurs after cessation.

Lastly, there is limited research on how fungus, yeast, and other components of the microbiome influences blood glucose control. The fungal and yeast components of the microbiome, named the mycobiome, has similarly been implicated in chronic inflammatory conditions [64]. Therefore, further research on this segment of the microbiome is warranted to understand if it contributes to blood glucose regulation. These limitations highlight the necessity for more robust, long-term clinical trials to assess microbiome-targeted therapies and their efficacy, safety, and practicality in managing T2Dm and obesity.

## 5. Conclusions

Utilizing the same pathways as traditional pharmacology, our gut’s microbiome has significant impacts on our body’s regulation of blood glucose. The modulation of incretins can provide significant benefits to patients living with type 2 diabetes mellitus or obesity. As emerging research has identified several strains of bacteria with favorable effects on this system, further human clinical trials should investigate these strains for their efficacy and long-term impact on health. Furthermore, the impacts of yeast and fungus remain unexplored. Perhaps more importantly, the functional testing of the gut needs significant growth in validation and future clinical studies will help to better refine the current approach to functional predictions of the gut microbiome. As future research in this field further develops our understanding, a probiotic approach to weight and blood glucose management, in conjunction with healthy dietary habits and regular exercise, may provide an effective management solution for patients who want to manage their condition without the addition of medications.

## Figures and Tables

**Figure 1 biomedicines-12-02719-f001:**
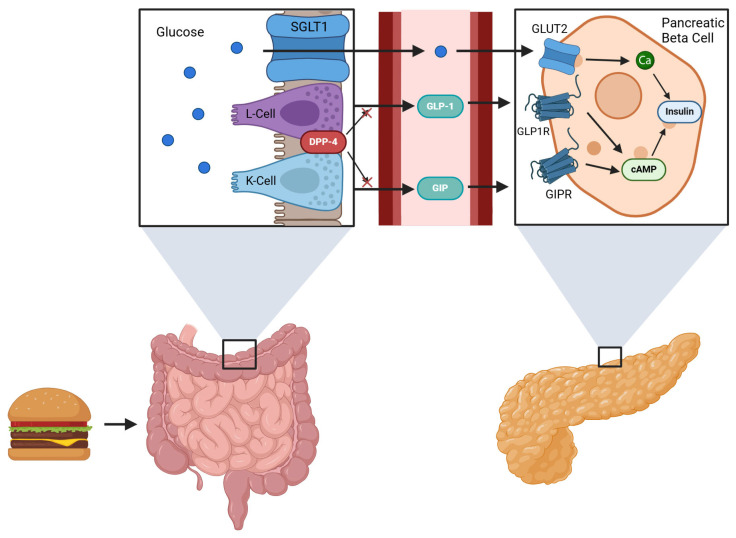
Nutrients from food intake stimulate the L-cells to produce GLP-1 and K-cells to produce GIP. The incretins stimulate insulin release in the pancreatic beta-cell via cAMP. DPP-4 metabolizes these incretins to prevent insulin production. Glucose enters the blood stream via SGLT1 (sodium-glucose cotransporter 1) and stimulates insulin release in the beta cell via GLUT2 (glucose transporter 2) [20,21,22].

**Figure 2 biomedicines-12-02719-f002:**
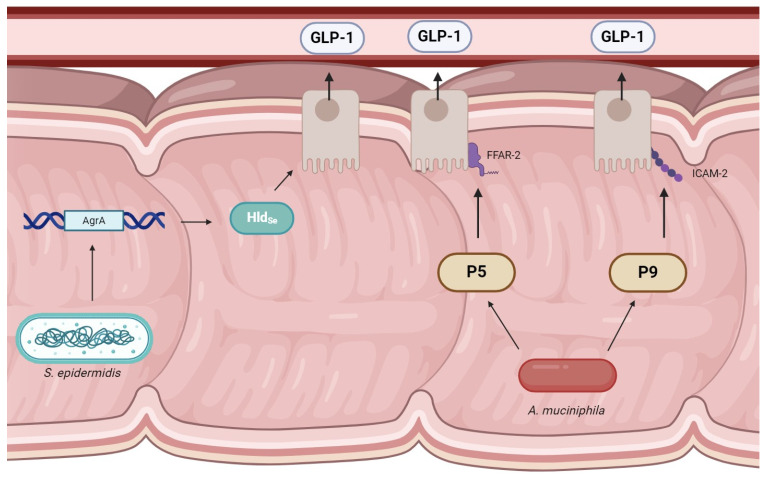
Within the intestines, *S. epidermidis* strains, which exhibit the AgrA gene can produce Hld_Se_, which has demonstrated stimulatory effects on GLP-1. Additionally, *A. mucinphila* produces P5 and P9 peptides. The P5 protein is thought to act on the FFAR-2 receptor on L-type cells, and P9 is hypothesized to interact with ICAM-2, which is located in the microvilli of endothelial cells. GLP-1 is released into the surrounding blood vessels, which then act on the pancreatic beta cells to produce insulin. The mechanism of action of *Bifidobacterium animalis* subsp. *lactis* MN-Gup is not currently well described in the literature.

**Table 1 biomedicines-12-02719-t001:** In vivo studies investigating prebiotic, probiotic, and postbiotic supplementation to modulate insulin, incretins, and lipids.

Study	Intervention	Dose/Day	Host	Significant Findings (*p*-Value)
1 [33]	*Bifidobacterium animalis* subsp. *lactis*	1 × 10^10^ CFU/kg BW	T2DM Mice	FBG: −3.83 mmol/L (0.001), Increased acetate (0.05), GLP-1 (0.05)
1 [33]	*Bifidobacterium animalis* subsp. *lactis*	2 × 10^9^ CFU/kg BW	T2DM Mice	FBG: −3.13 mmol/L (0.001), increased acetate (0.05), GLP-1 (0.05)
2 [34]	*Bifidobacterium animalis* subsp. *lactis*	1 × 10^10^ CFU/kg BW	Mice	Reduced weight gain (<0.05), reduced TC, TG, LDL (0.05)
2 [34]	*Bifidobacterium animalis* subsp. *lactis*	2 × 10^9^ CFU/kg BW	Mice	Reduced TC, TG, LDL (0.05)
3 [35]	*Bifidobacterium animalis* subsp. *lactis* and fermented milk	1 × 10^8^ CFU/g of fermented milk	Obese Mice	Reduced TC, TG, LDL (0.05)
4 [36]	*S. epidermidis* JA1	2 × 10^8^ cells	Obese Mice	−5.5% BW (0.05), −7% food intake (0.0082), −83.7% adiposity (<0.0001), −36% fasting insulin (0.0004)
5 [37]	*Bifidobacterium animalis* ssp. *lactis* 420, PDX	10^9^ CFU	T2DM Mice	B420 reduced glucose tolerance (0.002), PDX can stimulate GLP-1
6 [38]	Vancomycin, Bacitracin	Not reported	Obese Mice	2.03-fold increase in active GLP-1; reduced glucose tolerance, hyperinsulinemia, abundance of Firmicutes and Bacteroidetes
7 [39]	P9 Protein from *A. muciniphila*	100 μg per mouse	Obese Mice	Stimulation of GLP-1 secretion via ICAM-2
8 [40]	Maltose and Miglitol	2 g/kg, 10 mg/kg	Mice	Stimulation of GLP-1, Inhibition of GIP

CFU, colony-forming unit; BW, body weight, T2DM, type 2 diabetes mellitus, FBG, fasting blood glucose; GLP-1, glucagon-like peptide-1; TC, total cholesterol; TG, triglycerides: LDL, low-density lipoprotein; PDX, prebiotic polydextrose; ICAM-2, intercellular adhesion molecule 2; GIP, glucose-dependent insulinotropic polypeptide.

**Table 2 biomedicines-12-02719-t002:** In vitro studies investigating bacterial impact on incretin production.

Study	Bacteria	Signficant Findings
1	*Lactobacillus*, *Bifidobacterium*	CFE of BB12, L-8, 6-2, and B-53 demonstrated the most potent α-glucosidase inhibitory activity CFE of LGG, L-8, and B-84 demonstrated the highest DPP-IV inhibitory activity
2	*S. epidermidis*	JA1 and JA8 demonstrated the strongest GLP-1 stimulatory effects *AgrA* is responsible for activiting the Hld_Se_ peptide to stimulate GLP-1
3	*A. muciniphila*	P5 protein stimulates GLP-1 via the free fatty acid receptor on STC-1 cells
4	*A. muciniphila*	P9 protein upregulates the GLP-1 biosynthesis genes GCG and PCSK1

CFE, cell-free extract.

## Data Availability

No new data were generated from this report.

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
