# Peer review of "The Gut Microbiome’s Influence on Incretins and Impact on Blood Glucose Control"

_biomedicines, 2024, doi:10.3390/biomedicines12122719_

Round 1
Reviewer 1 Report
Comments and Suggestions for Authors
This is an interesting paper. However, there is some question regarding the methodology used. From what has been described in the methods section, it is difficult to fully understand what type of review this is. It is written like a systematic review, but PRISMA, or equivalent, is missing for it to be a systematic or scoping review. The inclusion and exclusion criteria are not well defined. These need to more specific. Also think about the paper exclusion policy you employed. Whilst you say there wasn't one, the fact you only used newer paper suggests, in fact, one was employed.
Why did you use Google Scholar and Pubmed? Did you exclusion criteria take into the fact that Google Scholar also includes preprints in the search? I.e. did you only survey peer reviewed literature?
The table is a nice inclusion, but it shows 8 different in vivo studies, but in the results section that you included 11 papers, 4 of which were in vitro studies. This would seem to suggest there should only be 7 in vivo studies included. Also the current formatting makes the table difficult to read.
It would be nice to see a table comparing in vitro studies.
Sub-headings in the results section would help understand when you are discussing in vitro or in vivo assays.
Make sure that bacterial names are consistently italicised (see table 1 in particular) and follow correct naming convention throughout.
"LLactobacillus" on line 174
Comments on the Quality of English LanguageEnglish is fine.
Author Response
Response to Reviewer 1 Comments
Point 1: “This is an interesting paper. However, there is some question regarding the methodology used. From what has been described in the methods section, it is difficult to fully understand what type of review this is. It is written like a systematic review, but PRISMA, or equivalent, is missing for it to be a systematic or scoping review. The inclusion and exclusion criteria are not well defined. These need to more specific. Also think about the paper exclusion policy you employed. Whilst you say there wasn't one, the fact you only used newer paper suggests, in fact, one was employed.”
Response 1: Thank you for your feedback. This paper intends to be a narrative review and thus we have reorganized the paper to follow the organization for reviews published by Biomedicines. We did not have an exclusion based on year of publication; as this is an emerging topic in research much of the published literature is within the last 3 years. We mention this in the text and edited it to acknowledge the recent growth in publications: “The review process did not limit articles based on publication date but resulted in more recent studies, reflecting the emerging developments in the fields.”
Point 2: “Why did you use Google Scholar and Pubmed? Did you exclusion criteria take into the fact that Google Scholar also includes preprints in the search? I.e. did you only survey peer reviewed literature?”
Response 2: We used Google Scholar in conjunction with Pubmed to aid in retrieving articles which are published to journals which are not indexed on Pubmed. Preprints were not included. We have included this phrase in the exclusion criteria which clarifies that all articles included were peer-reviewed and published: “Exclusion criteria included studies that lacked clear relevance to microbiome modulation or , were solely focused on unrelated metabolic pathways, were not published in English, or was not published in a peer reviewed journal”.
Point 3: The table is a nice inclusion, but it shows 8 different in vivo studies, but in the results section that you included 11 papers, 4 of which were in vitro studies. This would seem to suggest there should only be 7 in vivo studies included. Also the current formatting makes the table difficult to read.
Response 3: We have corrected the text to demonstrate that a total of 12 articles were included in the review. Four of which were in-vitro and the remaining eight were in-vivo articles. We have adjusted the formatting of the table to be left-justified to improve the readability of the paper.
Point 4: It would be nice to see a table comparing in vitro studies.
Response 4: We have incorporated this feedback in Table 2
Point 5: Sub-headings in the results section would help understand when you are discussing in vitro or in vivo assays.
Response 5: We have organized the sections of the paper by bacteria to provide further organization. Within each subsection of bacteria we then discuss the in vitro and in vivo included papers.
Point 6: Make sure that bacterial names are consistently italicised (see table 1 in particular) and follow correct naming convention throughout.
Response 6: We have corrected the font of bacterial species to ensure correct italics.
Point 7: "LLactobacillus" on line 174
Response 7: This spelling has been corrected, thank you.
Reviewer 2 Report
Comments and Suggestions for Authors
The authors aim to provide an overview of how gut microbiome-targeted approaches can improve glycemic control and reduce the risk of diabetes-related complications.
I have some suggestions:
1) line 64: the authors can delve deeper into the concept of diabesity and how the element that links the physiopathology of T2D and obesity is insulin resistance, in which the gut microbiota also plays a role (Microbiota-gut-brain axis: relationships among the vagus nerve, gut microbiota, obesity, and diabetes. Acta Diabetol. 2023 Aug;60(8):1007-1017. doi: 10.1007/s00592-023-02088-x.)
2) Figure 1, while informative, could benefit from a more detailed representation of the physiological process of incretin stimulation. I encourage the authors to enrich it by showing the cellular processes more specifically.
3) Line 99: Given the recent studies conducted on the dual agonist GIP/GLP-1, I ask the authors to include in the discussion other studies supporting how this drug's dual incretin stimulation is effective in glycemic control and weight reduction (SURMOUNT-1, doi: 10.1056/NEJMoa2206038; The SURMOUNT-4 Randomized Clinical Trial. JAMA 2024 Jan 2;331(1):38-48. Doi: 10.1001/jama.2023.24945; Tirzepatide's innovative applications in the management of type 2 diabetes and its future prospects in cardiovascular health. Front Pharmacol. 2024 Aug 28;15:1453825. doi: 10.3389/fphar.2024.1453825.)
4) I suggest to insert a flow chart that summarizes the process of inclusion and exclusion of the papers considered
5) there is a typo on line 123.
6) In the discussion, it is crucial to include a mention of how the effect of Metformin therapy on glycemic control is also mediated by its action on the gut microbiota (Therapeutic potential of butyrate for treatment of type 2 diabetes. Front Endocrinol. 2021;12:761834. https://doi.org/10.3389/fendo.2021.761834. ; Gut Microbiome, Microbial Metabolites and
Cardiometabolic Risk, Endocrinology, https://doi.org/10.1007/978-3-031-08115-6_8-1)
Author Response
Response to Reviewer 2 Comments
Point 1: “line 64: the authors can delve deeper into the concept of diabesity and how the element that links the physiopathology of T2D and obesity is insulin resistance, in which the gut microbiota also plays a role (Microbiota-gut-brain axis: relationships among the vagus nerve, gut microbiota, obesity, and diabetes. Acta Diabetol. 2023 Aug;60(8):1007-1017. doi: 10.1007/s00592-023-02088-x.)
Response 1: We have included a sentence which further explains the underlying mechanisms between diabetes and obesity: “Insulin resistance is a common underlying feature of T2DM and obesity which manifests from decreased insulin-stimulated glucose secretion and metabolism, and impaired suppression of hepatic glucose production as well as contribution from the gut microbiota”
Point 2: Figure 1, while informative, could benefit from a more detailed representation of the physiological process of incretin stimulation. I encourage the authors to enrich it by showing the cellular processes more specifically.
Response 2: The underlying cellular processes in which the bacteria induce GLP-1 secretion is poorly understood at the moment. We believe that further discussion and representation of this is outside of the scope of this manuscript as we don’t have the full scientific understanding as of yet but we appreciate the reviewer’s suggestion.
Point 3: Line 99: Given the recent studies conducted on the dual agonist GIP/GLP-1, I ask the authors to include in the discussion other studies supporting how this drug's dual incretin stimulation is effective in glycemic control and weight reduction (SURMOUNT-1, doi: 10.1056/NEJMoa2206038; The SURMOUNT-4 Randomized Clinical Trial. JAMA 2024 Jan 2;331(1):38-48. Doi: 10.1001/jama.2023.24945; Tirzepatide's innovative applications in the management of type 2 diabetes and its future prospects in cardiovascular health. Front Pharmacol. 2024 Aug 28;15:1453825. doi: 10.3389/fphar.2024.1453825.)
Response 3: We have included the following text which further elaborates on the effects of tirzepatide: “Dual agonists offer enhanced glucose control and more significant weight loss due to the additive effects of GIP and GLP-1. In clinical trials, tirzepatide has led to more than 10% body weight loss, and reductions in HbA1c (maximum reduction of 2.24%). Tirzepatide is also currently being investigated for its benefits on cardiovascular event risk.”
Point 4: I suggest to insert a flow chart that summarizes the process of inclusion and exclusion of the papers considered
Response 4: As we have identified this paper as a narrative review we have further elaborated on the search strategy and do not believe that a flow chart will be of the same benefit that it could provide to a systematic review.
Point 5: there is a typo on line 123.
Response 5: We have corrected this.
Point 6: In the discussion, it is crucial to include a mention of how the effect of Metformin therapy on glycemic control is also mediated by its action on the gut microbiota (Therapeutic potential of butyrate for treatment of type 2 diabetes. Front Endocrinol. 2021;12:761834. https://doi.org/10.3389/fendo.2021.761834. ; Gut Microbiome, Microbial Metabolites and
Cardiometabolic Risk, Endocrinology, https://doi.org/10.1007/978-3-031-08115-6_8-1)
Response 6: We have included discussion and referenced papers which have shown that metformin treatment can increase the abundance of Akkermansia: “. Furthermore, research into metformin has shown that it increases the abundance of A. muciniphila in diet-induced obese mice and humans40. While in-vitro studies have identified specific proteins from Akkermansia which can aid in insulin response and human trials have demonstrated associations between Akkermansia and metabolic changes, clinical trials are needed to demonstrate that Akkermansia supplementation can induce incretin stimulation in animal models and humans.”
Reviewer 3 Report
Comments and Suggestions for Authors
The manuscript titled "The Gut Microbiome Impact on Incretins and Blood Glucose Control" presents an interesting topic; however, the scope and depth of the review are limited. The manuscript only analyzes 11 articles, which restricts the breadth of discussion and the information provided to readers. The review lacks comprehensive coverage, especially in the areas of prebiotics, probiotics, and postbiotics. In its current form, the manuscript does not provide enough comprehensive information or critical analysis to justify publication. I recommend a major revision, addressing the limitations outlined below.
Major concerns:
The discussion primarily focuses on common bacterial genera like Bifidobacterium, Lactobacillus, and Akkermansia. While these microbes are indeed relevant to blood glucose regulation, the review would benefit from including additional microbial strains. Expanding the scope to incorporate research on other beneficial bacteria would provide more depth and balance. Moreover, the section on postbiotics is underdeveloped. The review should elaborate on postbiotics, including their mechanisms of action. A schematic diagram illustrating the mechanisms of various gut microbiota and their metabolites on incretins and glucose metabolism would greatly enhance clarity and understanding.
The review should also discuss nutritional interventions that modulate the gut microbiome and their potential role in regulating blood glucose levels. This would offer clinical relevance, providing practitioners with actionable insights into how microbiome modulation can be applied for better glucose control.
The manuscript adopts a structure more suited to a research article, with sections like methods, results, discussion, and conclusion. As a review paper, it would be more appropriate to organize the content under sections such as background, literature review, current understanding, gaps, and future perspectives.
Integrate the latest research findings, especially from the past 3 years, to ensure the review reflects current scientific advancements in the field. Highlight any new insights or breakthroughs that have emerged recently.
Provide detailed information regarding the methodology, specifically outlining the time frame during which the research articles were collected. Specify the range of years covered in the literature review to give readers a clear understanding of its scope.
Table 1. In addition to animal model studies, the review should incorporate recent clinical studies. This inclusion will strengthen the review by providing evidence of how gut microbiome interventions affect human blood glucose control, thereby enhancing its applicability to clinical practice.
Author Response
Response to Reviewer 3 Comments
Point 1: The manuscript titled "The Gut Microbiome Impact on Incretins and Blood Glucose Control" presents an interesting topic; however, the scope and depth of the review are limited. The manuscript only analyzes 11 articles, which restricts the breadth of discussion and the information provided to readers. The review lacks comprehensive coverage, especially in the areas of prebiotics, probiotics, and postbiotics. In its current form, the manuscript does not provide enough comprehensive information or critical analysis to justify publication. I recommend a major revision, addressing the limitations outlined below.
Major concerns:
The discussion primarily focuses on common bacterial genera like Bifidobacterium, Lactobacillus, and Akkermansia. While these microbes are indeed relevant to blood glucose regulation, the review would benefit from including additional microbial strains. Expanding the scope to incorporate research on other beneficial bacteria would provide more depth and balance. Moreover, the section on postbiotics is underdeveloped. The review should elaborate on postbiotics, including their mechanisms of action. A schematic diagram illustrating the mechanisms of various gut microbiota and their metabolites on incretins and glucose metabolism would greatly enhance clarity and understanding.
Response 1: We have clarified our research topic in focusing on the impact that the microbiome have on incretion production. In the state of current research into this topic, primarily Bifidobacterium, Lactbacillus, and Akkermansia have been identified to modulate incretin production.
Point 2: The review should also discuss nutritional interventions that modulate the gut microbiome and their potential role in regulating blood glucose levels. This would offer clinical relevance, providing practitioners with actionable insights into how microbiome modulation can be applied for better glucose control.
Response 2: We believe that, while interesting, nutritional interventions are outside of the scope of this current manuscript as we intend to focus on specific bacteria and bacterial interventions which can modulate incretin glucose control.
Point 3: The manuscript adopts a structure more suited to a research article, with sections like methods, results, discussion, and conclusion. As a review paper, it would be more appropriate to organize the content under sections such as background, literature review, current understanding, gaps, and future perspectives
Response 3: We appreciate the advice on organization and we have reorganized this manuscript to match the organization of the reviews submitted to Biomedicines
Point 4: Integrate the latest research findings, especially from the past 3 years, to ensure the review reflects current scientific advancements in the field. Highlight any new insights or breakthroughs that have emerged recently.
Response 4: Our search included studies that were within the past 3 years as well. We believe that our manuscript does include the latest research on this topic and helps identify important strains of bacteria which have incretin-modulating capabilities. As reviewer 1 highlighted, all of our included papers are within the last 3 years which demonstrates that this is an emerging field of research.
Point 5: Provide detailed information regarding the methodology, specifically outlining the time frame during which the research articles were collected. Specify the range of years covered in the literature review to give readers a clear understanding of its scope.
Response 5: We have added details on the range of years for the literature review.
Point 6: Table 1. In addition to animal model studies, the review should incorporate recent clinical studies. This inclusion will strengthen the review by providing evidence of how gut microbiome interventions affect human blood glucose control, thereby enhancing its applicability to clinical practice.
Response 6: We did not identify any human clinical trials which demonstrated modulation of the incretins after supplementation with bacteria.
Reviewer 4 Report
Comments and Suggestions for Authors
The gut microbiome is the totality of microorganisms, bacteria, viruses, protozoa, and fungi, and their collective genetic material present in the gastrointestinal tract. Probiotics and prebiotics are among the gut microbiota targeted approaches in the management of different diseases.
The main purpose of the current review is very confusing. If the goal is to clarify the effect of the gut microbiome (bacteria, viruses, protozoa, and fungi) on the level of blood glucose, as indicated by the title of the manuscript, then the authors must re-edit the manuscript in a manner appropriate to that, taking into account the effect of each component of the microbiome on the regulation of blood glucose and incritins, as well as the relationship between each bacterial phylum and its regulation of blood glucose and incritins.
The authors are advised to review the following articles.
Gérard, C., & Vidal, H. (2019). Impact of Gut Microbiota on Host Glycemic Control. Frontiers in endocrinology, 10, 29. https://doi.org/10.3389/fendo.2019.00029
Palmnäs-Bédard, M. S. A., Costabile, G., Vetrani, C., Åberg, S., Hjalmarsson, Y., Dicksved, J., Riccardi, G., & Landberg, R. (2022). The human gut microbiota and glucose metabolism: a scoping review of key bacteria and the potential role of SCFAs. The American journal of clinical nutrition, 116(4), 862–874. https://doi.org/10.1093/ajcn/nqac217
Angelini, G., Russo, S., & Mingrone, G. (2024). Incretin hormones, obesity and gut microbiota. Peptides, 178, 171216. https://doi.org/10.1016/j.peptides.2024.171216
Contrary to the title, the introduction and results in the current manuscript address the impact of probiotics and prebiotics as approaches targeting the gut microbiota in diabetes management. If this is the main point of the current review, then the authors should reorganize the manuscript with clarification of the role of probiotics and prebiotics in regulating diabetes, the mechanism of action of each of them, and their role in reducing dysbiosis, which is reflected in the management of diabetes.
Author Response
Response to Reviewer 4 Comments
Point 1: The gut microbiome is the totality of microorganisms, bacteria, viruses, protozoa, and fungi, and their collective genetic material present in the gastrointestinal tract. Probiotics and prebiotics are among the gut microbiota targeted approaches in the management of different diseases.
Response 1: We have pointed out that there is a limited amount of research into how fungus, yeasts, and other components of the microbiome can impact incretins. We only identified published research focusing on bacteria in this search. “Lastly, there is limited research on how fungus, and yeast, and other components of the microbiome influences blood glucose control. The fungal and yeast components of the microbiome, named the mycobiome, has similarly been implicated in chronic inflammatory conditions. Therefore, further research on this segment of the micro-biome is warranted to understand if it contributes to blood glucose regulation.”
Point 2: The main purpose of the current review is very confusing. If the goal is to clarify the effect of the gut microbiome (bacteria, viruses, protozoa, and fungi) on the level of blood glucose, as indicated by the title of the manuscript, then the authors must re-edit the manuscript in a manner appropriate to that, taking into account the effect of each component of the microbiome on the regulation of blood glucose and incritins, as well as the relationship between each bacterial phylum and its regulation of blood glucose and incritins.
The authors are advised to review the following articles.
Gérard, C., & Vidal, H. (2019). Impact of Gut Microbiota on Host Glycemic Control. Frontiers in endocrinology, 10, 29. https://doi.org/10.3389/fendo.2019.00029
Palmnäs-Bédard, M. S. A., Costabile, G., Vetrani, C., Åberg, S., Hjalmarsson, Y., Dicksved, J., Riccardi, G., & Landberg, R. (2022). The human gut microbiota and glucose metabolism: a scoping review of key bacteria and the potential role of SCFAs. The American journal of clinical nutrition, 116(4), 862–874. https://doi.org/10.1093/ajcn/nqac217
Angelini, G., Russo, S., & Mingrone, G. (2024). Incretin hormones, obesity and gut microbiota. Peptides, 178, 171216. https://doi.org/10.1016/j.peptides.2024.171216
Contrary to the title, the introduction and results in the current manuscript address the impact of probiotics and prebiotics as approaches targeting the gut microbiota in diabetes management. If this is the main point of the current review, then the authors should reorganize the manuscript with clarification of the role of probiotics and prebiotics in regulating diabetes, the mechanism of action of each of them, and their role in reducing dysbiosis, which is reflected in the management of diabetes.
Response 2: Thank you for this insight. Our manuscript is focused on the microbiome’s impact on incretin production and secondarily on how this affects glucose control. This has been demonstrated through research which measures incretin stimulation following the probiotic bacterial supplementation. This is not a manuscript about diabetes speficially except that diabetes includes glucose management as a central part of the disease. Therefore, we have adjusted the title to more accurately represent our intention.
Round 2
Reviewer 2 Report
Comments and Suggestions for Authors
The authors have partially responded to the comments previously suggested.
I still find it useful:
1) Figure 1, while informative, could benefit from a more detailed representation of the physiological process of incretin stimulation. I encourage the authors to enrich it by showing the cellular processes more specifically.
2) I suggest to insert a flow chart that summarizes the process of inclusion and exclusion of the papers considered.
Author Response
The authors have partially responded to the comments previously suggested.
I still find it useful:
Comment 1: Figure 1, while informative, could benefit from a more detailed representation of the physiological process of incretin stimulation. I encourage the authors to enrich it by showing the cellular processes more specifically.
Response 1 : Thank you for this feedback. We have revised the figure to show this process in more depth.
Comment 2: I suggest to insert a flow chart that summarizes the process of inclusion and exclusion of the papers considered. (This reviewer asked for this before but we'll continue to hold the line but I'll be much more direct in how I word my response. Simply state that we have already included the search criteria and we can state if we allowed human and animal studies but that's it and i'll add the rest).
Response 2: We appreciate the reviewer’s advice but we respectfully disagree. This is not a systematic review and the inclusion of a flow diagram will give it an appearance of a systematic review. We would like to emphasize that this is not a systemic review and therefore an inclusion/exclusion criteria flow diagram is not relevant except to state that all studies that include animal and clinical studies were included. We have already indicated the details of how we approached the review: “… inclusion criteria were broad to capture a wide scope of evidence. We included in vitro studies, as these provided mechanistic insights for future clinical applications. We also include in vivo studies, primarily focusing on clinical trials conducted in humans or animal models, particularly mice, given their widespread use in metabolic research…” We believe that this is sufficient.
Reviewer 4 Report
Comments and Suggestions for Authors
The authors made a clear effort in reviewing the manuscript.
Finally, please check the numbering according to the correct order of titles and subtitles in the manuscript.
Author Response
Comment 1: The authors made a clear effort in reviewing the manuscript. Finally, please check the numbering according to the correct order of titles and subtitles in the manuscript.
Response 1 : Thank you for this input we have corrected the numbering of the conclusion.